# Construct a Regional Innovation Ecosystem: A Case Study of the Beijing-Tianjin-Hebei Region in China

**Yuan Cao [1], Jingxian Liu [2], Ying Yang [1], Xiaolin Liu [1], Zhixuan Liu [2], Ning Lv [2], Hongkun Ma [2,3,\*], Zhenyao Wang [1] and Hongtu Li [1]**

[1] College of Geography and Environment, Shandong Normal University, Jinan 250358, China
[2] School of Economics, Shandong Normal University, Jinan 250358, China
[3] Academy for County Economy Research, Shandong Normal University, Jinan 250358, China
[\*] Correspondence: mahongkun@sdnu.edu.cn; Tel.: +86-15315114087; Fax: +86-0531-86181060

**Abstract:** Constructing a Regional Innovation Ecosystem (RIE) can be an effective way to utilize innovation resources by breaking the existing regional barriers. Existing research focuses more on exploring the characteristics, evolution, and impact of innovation ecosystems from a theoretical perspective, while few studies combine reality with exploring how to build an effective innovation ecosystem. Given that the concept of an innovation ecosystem is inspired by natural ecosystems, using the method of comparative analysis, this article begins by analyzing the characteristics of tropical rainforests as the most stable and efficient natural ecosystem and deeply explores the essential features of an efficient RIE, including an innovative soft and hard environment, multi-level and differentiated producers, consumers and decomposers of innovative activities, and a virtuous cycle of innovation resources. By comparing Beijing-Tianjin-Hebei's current reality with these characteristics, this article systematically analyzes the advantages of the innovation environment, the further improvement of innovation resource aggregation and innovation chains, as well as the disadvantages of an unbalanced distribution of innovation resources, the leapfrogging transformation of scientific and technological achievements, low innovation levels of market-oriented innovation entities, and loss of innovative factors, such as talents. Based on this analysis, targeted suggestions to construct a RIE in Beijing-Tianjin-Hebei, including using both market and administrative approaches to allocating innovation factors, stimulating the innovation vitality of market-oriented innovation entities, taking multiple measures to enhance Hebei Province's capacity, building an innovative environment that is livable and business-friendly are proposed.

**Keywords:** Beijing-Tianjin-Hebei; Regional Innovation Ecosystem; collaborative innovation; innovation drive





## 1. Introduction

The concept of a Regional Innovation Ecosystem (RIE) has gained significant attention in the academic literature and policy arena in recent years [1,2]. Based on the inspiration of the natural ecosystem, the RIE was first introduced by Moore in 1993 as a framework for understanding the dynamics of innovation within a specific region [3]. The RIE can be considered analogous to a natural ecosystem, as both systems involve interactions between various elements to maintain a balance and sustain their functions [4,5]. Just as a natural ecosystem requires the presence of diverse species and the cycling of nutrients and energy, a RIE requires the interaction and cooperation of different actors and the flow of knowledge, capital, and resources [6].

The RIE is a complex and dynamic system that involves various actors, such as universities, research institutions, firms, and government, and their interactions [7]. It aims to facilitate the creation and diffusion of knowledge, as well as the formation of new firms and industries [8]. Meanwhile, the RIE plays a critical role in strengthening technological

innovation, promoting regional economic growth, and enhancing the competitiveness of a region [9]. A well-functioning RIE can provide an environment that nurtures innovation by providing necessary resources, such as capital, talent, and infrastructure, and facilitating knowledge spillovers and collaborations among various actors [10]. Several studies have shown that the presence of a strong RIE is associated with higher levels of innovation and economic performance [11–14]. Renowned RIEs in the United States, European Union, and Canada have significantly influenced innovation within their respective cities and regions [15]. Notably, Silicon Valley (US) [16], the Øresund Region (EU) [17], and the Waterloo-Toronto Corridor (Canada) have become global innovation hubs [18]. These RIEs have fostered economic growth, attracted skilled talent, and facilitated collaborations between academia, industry, and government [19]. While plenty of existing research focusing more on exploring the characteristics, evolution, and impact of innovation ecosystems from a theoretical perspective, few studies combine reality and related theory to explore how to build an effective innovation ecosystem.

To address the existing limitations in the current research, this paper tries to combine reality and related theory to explore how to build an effective innovation ecosystem with the Beijing-Tianjin-Hebei region in China, as the case. The reason for choosing the Beijing-Tianjin-Hebei region in China as the case is as follows. Firstly, the Beijing-Tianjin-Hebei region holds the most concentrated scientific and technological innovation resources in China, and the establishment of a RIE in this region exhibits significant feasibility. Secondly, the development of a RIE in this region also possesses considerable practical significance. Currently, China's economic development is facing a complex and severe situation at home and abroad, with increasing contradictions and challenges [20]. To deal with these challenges, it is essential to maintain strategic stability and strengthen the innovation drive by pursuing the path of high-quality development [21]. Thirdly, the construction of a regional innovation ecosystem in the Beijing-Tianjin-Hebei area can serve as a reference and guide for enhancing innovation collaboration in other regions of China, as well as urban agglomerations around the world. Nowadays, with the significant improvement in urban development levels, the connectivity and communication between Chinese cities and other domestic and international cities have become increasingly active [22]. In the case of the Beijing-Tianjin-Hebei region, Beijing-based enterprises are rapidly raising their economic power [23], and Beijing is one of the leading cities in the Information Technology sector in the world [24], constructing a regional innovation ecosystem in the area not only further enhances the innovation synergy among relevant cities within the region, but also serves as a model for the innovation synergy in other cities.

Based on this, this study focuses on the Beijing-Tianjin-Hebei region as a case study, comparing the Regional Innovation Ecosystem (RIE) with a natural ecosystem to explore the characteristics of the RIE. The main structure of this article is as follows (Figure 1). The second part mainly introduces the methodology employed in this article. In the third part, the features of RIE are investigated. Then, the existing foundation and favorable conditions for the construction of the RIE in the Beijing-Tianjin-Hebei region are systematically sorted out in the fourth part of the paper. The fifth part of the paper identifies weaknesses and deficiencies faced by the Beijing-Tianjin-Hebei region in the construction of the RIE, still using the characteristics of a stable and efficient RIE as a reference. After being deeply discussed in the sixth part, this paper concludes with targeted policy recommendations in the seventh part.

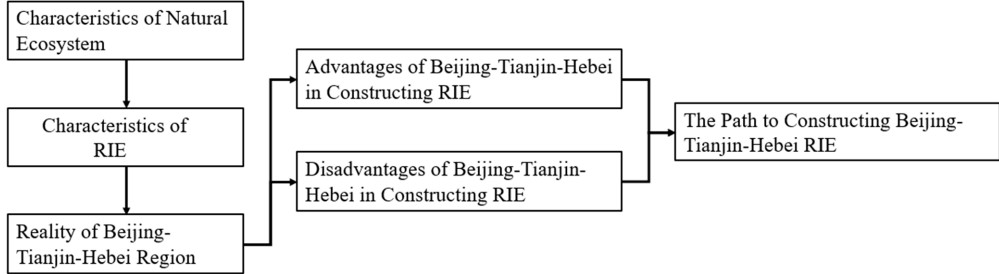

**Figure 1.** The research framework of this paper.

## 2. Methodology

### 2.1. Methods

Comparative analysis is employed in this article. Comparative analysis is a research method that involves identifying and examining similarities and differences between two or more objects of study [25,26]. It is commonly used in various fields, such as sociology, political science, and business, to explore and understand the complexities of social phenomena. Since the concept of RIE was first introduced by Moore based on the inspiration of natural ecosystems, there inevitably exists a significant degree of comparability between these two concepts. In this article, to construct the RIE, it is important to first understand what the characteristics of a stable and efficient RIE are. In this regard, using the comparative analysis method, this paper selects the tropical rain forest, the most stable and efficient natural ecosystem, as the analysis object to identify its characteristics and then uses this as a reference to analyze the characteristics of a stable and efficient RIE.

### 2.2. Study Area

To deeply explore how to construct a RIE, this paper employed the selection region of Beijing-Tianjin-Hebei of China as a case. The reason for choosing the region of Beijing-Tianjin-Hebei of China as the study case is as follows (Figure 2). Firstly, Beijing-Tianjin-Hebei of China is the region with the most concentrated scientific and technological innovation resources in China, which facilitates the study of the characteristics of RIE. Secondly, by constructing a RIE in the Beijing-Tianjin-Hebei region, it is advantageous to provide a model and demonstration for other regions in the country to facilitate the flow of innovation elements, such as talent, technology, capital, and information.

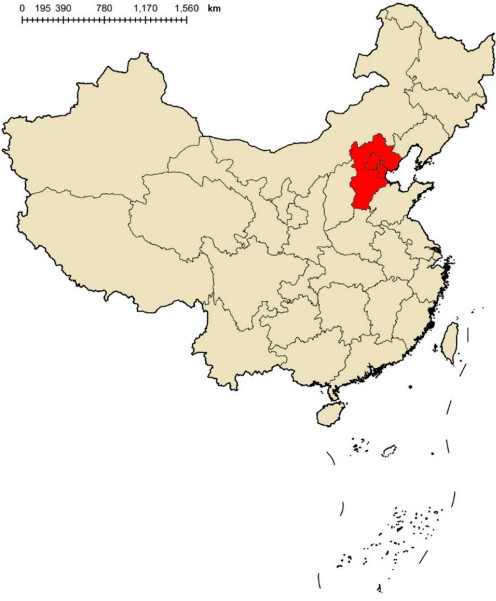

**Figure 2.** The selected study area of Beijing-Tianjin-Hebei. Note: The area with red color refers to Beijing-Tianjin-Hebei, including Beijing City, Tianjin City and Hebei Province.

*2.3. Data Sources*

The unique data in this study are sourced from Statistical Yearbook, Science and Technology Statistical Yearbook in the related years, Ministry of Education, Ministry of Science and Technology and Chinese Academy of Engineering. All the exact data sources are detailed and explained below in the related tables, respectively.

## 3. Regional Innovation Ecosystem and Tropical Rainforest

The concept of the innovation ecosystem is borrowed from the natural ecosystem, and to a certain extent, it can be said to be derived from the latter [3]. In an ideal situation, a stable and efficient innovation ecosystem and a stable and efficient natural ecosystem have great similarities in their composition structure, operation mechanism, and operation rules [27,28].

*3.1. Inherent Connections between Natural Ecosystems and Innovation Ecosystems*

The concept of an innovation ecosystem has evolved from the application of ecological principles to the study of innovation processes and regional development [3,29]. Several milestones can be identified in this evolution. Firstly, the biological analogy. Early works, such as Schumpeter's "creative destruction" (1942) [30] and Nelson and Winter's "evolutionary theory of economic change" (1982) [31], introduced biological metaphors to describe innovation processes and competition in economic systems. Secondly, the emergence of innovation systems. The innovation systems concept, first introduced by Freeman (1987) [32] and Lundvall (1992) [33], highlights the complex interactions among various actors, institutions, and resources that drive innovation in regional, national, and sectoral contexts. Thirdly, the adoption of ecological principles. Moore (1993) [3] introduced the concept of "business ecosystems," emphasizing the co-evolution, mutualism, and interdependence among firms and their environment. This marked the first explicit application of ecological principles to the study of innovation and economic systems. Fourthly, the development of innovation ecosystems. Building on the previous milestones, the "innovation ecosystem" concept emerged, emphasizing the dynamic and complex interactions among various actors, resources, and institutions that drive innovation, knowledge creation, and economic growth [34,35].

Despite their distinct focus, natural ecosystems and innovation ecosystems share several fundamental principles which underpin their inherent connections (Table 1). Firstly, interdependence. Both ecosystems rely on complex networks of relationships, where actors or species depend on each other for resources and support [29,36]. Secondly, co-evolution. Both ecosystems exhibit co-evolutionary processes, where actors or species adapt and evolve in response to changes in their environment [37]. Thirdly, resilience and adaptability. Both ecosystems possess the ability to recover from disturbances and adapt to changing conditions, highlighting the importance of diversity, redundancy, and flexibility [38,39].

**Table 1.** A comparison between natural ecosystems and regional innovation ecosystems.

| | Natural Ecosystem | RIE |
|---|---|---|
| External Environment | Suitable for Biological Growth, including appropriate temperature, abundant sunlight, humid air, and ample moisture, etc. | The government need to increase hardware and institutional supply to form a suitable environment for innovation activities. |
| Internal Structure | Complex, stable, and diverse structure with producers, consumers, and decomposers performing their respective roles. | In an innovation ecosystem, producers mainly include universities, research institutes, and enterprises engaged in innovation activities; consumers are composed of national needs and market-oriented demands for scientific and technological achievements. Decomposers refers to the promotion of supply-side structural reforms. |
| Element Cycling | A virtuous closed-loop cycle of material and energy | Innovation factors realize a virtuous closed-loop cycle within the innovation ecosystem. |

### 3.2. The Features of a Tropical Rainforest Ecosystem

In today's world, the tropical rainforest ecosystem is widely recognized as the most stable and metabolically efficient natural ecosystem [40,41]. By analyzing in-depth, the characteristics and rules of the tropical rainforest ecosystem in terms of its composition structure, operation mechanism, and operation rules, reflecting on the RIE from this perspective will undoubtedly provide an effective reference for building a stable and efficient RIE (Table 1).

(1) An External Environment Suitable for Biological Growth

Tropical rainforest ecosystems are generally distributed in tropical regions between 10 degrees north and south latitudes on both sides of the equator [42]. Due to its proximity to the equator, this region receives longer periods of light than any other region on Earth [43]. The equatorial climate in this region has the following three characteristics: first, high temperature, with an average annual temperature of 25–28 °C in the tropical rainforest area, and small temperature fluctuations, which is suitable for plant growth [44]; second, high rainfall, with normal annual rainfall in the tropical rainforest area ranging from 1750 to 2000 mm, and rainfall is relatively evenly distributed without distinct wet and dry seasons, allowing plants to obtain the necessary water for rapid growth throughout the year [45]; third, high humidity, with a small air pressure gradient and little airflow, weak or static winds, and water not easily lost [46]. Overall, the tropical rainforest region has an excellent and stable external environment for plants and animals, which is very conducive to their metabolism and growth.

(2) Complex, Stable, and Diverse Structure with Producers, Consumers, and Decomposers Performing Their Respective Roles

The tropical rainforest ecosystem has a complex and diverse composition structure [47,48]. In this ecosystem, there are not only towering trees up to 45–50 m in height but also shrubs and grasses growing close to the ground and various animals growing among them. In addition, there are microorganisms that grow actively underground and are difficult to see with the naked eye. These complex and diverse members in their respective ecological niches reproduce and thrive and are closely interconnected, making them an indispensable part of the tropical rainforest ecosystem [49,50]. From another perspective, thanks to the complex and diverse composition structure and the respective roles played by producers, consumers, and decomposers, the tropical rainforest has become the most stable forest ecosystem on Earth.

(3) Achieving a virtuous closed-loop cycle of material and energy

In the tropical rainforest ecosystem, material flow is circulated through the food chain and food web, ultimately forming a virtuous closed-loop cycle in the ecosystem [51]. Energy, which is attached to matter, is also circulated through the ecosystem by means of energy flow. Specifically, photosynthetic organisms, such as green plants, fix solar energy in the form of organic matter. Primary consumers, such as herbivores, digest and absorb the majority of the stored energy in plants and convert it into energy forms needed for their own growth and development, such as adenosine triphosphate (ATP) [52,53]. Secondary or higher-level consumers, such as carnivores, absorb the energy stored in lower-level consumers by consuming them. Decomposers break down organic matter in the bodies of consumers at all levels, releasing nutrients in inorganic form into the soil [1,54]. Green plants then absorb these nutrients, and the cycle of material and energy is closed in a virtuous loop. It should be noted that, although about 90% of energy is lost during the transfer of energy from lower to higher trophic levels, a virtuous closed-loop cycle of material and energy is achieved and ultimately becomes stable due to the sustained input of external energy, such as sunlight.

### 3.3. The Characteristics of RIE: Learning from the Tropical Rainforest

(1) Creating an innovation ecosystem environment conducive to the development and growth of innovation activities

Just as natural ecosystems require a suitable environment for survival, the construction of an innovation ecosystem also requires the government to increase hardware and institutional supply to form a suitable environment for innovation activities [55]. Hardware refers to the infrastructure, such as transportation, food, housing, office space, and various trading markets necessary for the daily operations of innovative talents and innovative entities. The soft environment consists of various policies, regional policies, industrial policies, and regulations that incentivize innovation [56,57].

(2) Multi-level, differentiated producers, consumers, and decomposers of innovation activities

Producers of innovation activities are located at different levels. In a natural ecosystem, producers produce carbohydrates, which become the food for consumers [58]. In an innovation ecosystem, producers mainly include universities, research institutes, and enterprises engaged in innovation activities. They produce innovative achievements, which become the "food" for national science and technology demand or science and technology achievement transformation institutions. A mature and efficient innovation ecosystem requires innovation entities with varying degrees of innovation, different geographical distributions, and complementary functional implementation [59,60]. This is reflected firstly in innovation entities with regional targeting according to their goals and innovation resource endowment. Additionally, there should be both large or flagship innovation entities aimed at cutting-edge technologies, as well as small and medium-sized innovation entities aimed at serving the needs of specific market segments.

An improved mechanism for the transformation of scientific and technological achievements is also required [61,62]. In a natural ecosystem, consumers eat carbohydrates produced by producers. In an innovation ecosystem, consumers are composed of national needs and market-oriented demands for scientific and technological achievements. An efficient innovation ecosystem must emphasize the effective connection between scientific and technological innovation and national needs while building an improved mechanism for the market-oriented transformation of scientific and technological achievements. This includes encouraging the development of angel investment, venture capital, private equity, and other institutions and platforms involved in the transformation of scientific and technological achievements.

Accelerating the promotion of supply-side structural reforms is also important. In a natural ecosystem, decomposers release resources back into the ecosystem by decomposing animal and plant remains and food residues. An efficient innovation ecosystem also requires such "decomposers". By accelerating the promotion of supply-side structural reforms, market entities with low innovation capabilities and difficulties in transformation can orderly exit their operations, releasing scarce resources, such as labor, subsidies, and finance, back into the innovation ecosystem to support the high development of the overall capacity of innovation.

(3) Innovation factors realize a virtuous closed-loop cycle within the innovation ecosystem

Likewise, an efficient and stable innovation ecosystem should also realize the benign cycling of various innovative factors [63]. By improving institutional mechanisms, all kinds of innovative entities can enhance research and development efficiency after obtaining an effective supply of key innovative elements, such as talent and funds [64]. Additionally, while strengthening the application orientation of innovation activities of innovative entities, improving their ability to serve national strategies and market demands, various market-oriented result transformation organizations should be encouraged to actively promote the commercialization of scientific and technological achievements, thereby creating more wealth in the critical stage of commercialization. Furthermore, by accelerating supply-side structural reform, a large number of factors inefficiently occupied can be released in an orderly manner, together with the wealth created by the transformation of scientific and technological achievements and become the food for the next round of innovation activities, thereby changing the one-way input-output model and achieving the benign cycling of innovation factors (Table 1).

## 4. Conditions for Constructing a RIE in a Combined Region of Beijing-Tianjin-Hebei

Based on the analysis of the characteristics of the RIE in the previous sections, this section will outline the existing conditions for constructing a RIE in the Beijing-Tianjin-Hebei area. It is found that the innovation environment in the Beijing-Tianjin-Hebei area is relatively superior, with high aggregation of innovative factors and a further improvement of the innovation chain from original innovation to outcome transformation, which constitutes favorable conditions for constructing a RIE in the area.

### 4.1. Possessing Suitable Hard and Soft Environments for the Development and Growth of Innovation Activities

In terms of hard environments, the Beijing-Tianjin-Hebei area is located in the North China Plain and adjacent to the Bohai Sea and has formed a three-dimensional transportation system of land, sea, and air, making it one of the regions with the complete hardware infrastructure in China. In terms of land and sea transportation, the Beijing-Tianjin-Hebei area has four ports, with a railway operating mileage of about 3.4 times the national average, and high-speed railways covering nearly 80% of cities at and above the prefecture level, with an average density of expressways about 3.1 times the national level, which has basically achieved a one-hour traffic circle. In terms of air transportation, the Beijing-Tianjin-Hebei area currently has a group of nine airports. The complete transportation network will improve the communication efficiency between the various subjects in the Beijing-Tianjin-Hebei area and with external subjects, reduce communication costs, and provide convenience for innovation activities.

In recent years, the national and local governments in the Beijing-Tianjin-Hebei area have conducted a large number of institutional innovations to support the improvement of the innovation level in the region and initially formed a soft environment that supports and encourages innovation. At the national level, the "Outline of the Plan for the Coordinated Development of the Beijing-Tianjin-Hebei Region" was issued in 2015, which serves as a top-level design for the integration of regional resources and the implementation of innovation-driven development. At the local government level, the governments of the three cities in the Beijing-Tianjin-Hebei area have successively issued a series of guiding or normative documents, such as "Opinions on Strengthening the Construction of Key Platforms for Industrial Transfer and Undertaking in the Beijing-Tianjin-Hebei Region", "Action Plan for Building the Zhongguancun National Independent Innovation Demonstration Zone's Joint Innovation Community in the Beijing-Tianjin-Hebei Region (2016–2018)", "Implementation Opinions on Distributing Policies Oriented Toward Increasing Knowledge Value", and "Three-year Action Plan for Science and Technology Innovation in Hebei Province (2018–2020)". The issuance of these guiding or normative documents has established a policy framework and provided institutional guidance for the construction of a RIE in the Beijing-Tianjin-Hebei area, thus becoming an important soft environment that has a significant impact on the RIE.

Moreover, with only 2.3% of the national land area and 8% of the population, the Beijing-Tianjin-Hebei area has created 10% of the national GDP. The GDP of Beijing and Tianjin alone has exceeded one trillion yuan. Economic development has ensured fiscal revenue and provided strong public funding support for scientific and technological innovation. At the same time, the vitality of economic development and the industrial foundation of the developed economy are also drivers that promote the improvement of the scientific and technological innovation level in the Beijing-Tianjin-Hebei area.

### 4.2. Having a Certain Degree of Innovation Subject and Innovation Element Aggregation

The Beijing-Tianjin-Hebei region is currently the most concentrated area for various types of innovative entities, talents, funds, and other innovation elements in China. In terms of innovative entities, the number of universities, research and development institutions, and innovative enterprises in the Beijing-Tianjin-Hebei region ranks among the top in the country. In particular, institutions representing top-level basic and applied research

capabilities, such as first-class universities, first-class disciplines, national key laboratories, and national engineering technology research centers, show significant advantages in the Beijing-Tianjin-Hebei region. At the end of 2021, the region accounted for 23.81% of the country's first-class universities, 48.97% of the first-class disciplines, 33.86% of the national key laboratories, and 22.22% of the national engineering research centers (Table A1). A large number of first-class research subjects form research clusters with strong scale effects and spillover effects and have huge potential for collaborative innovation. In terms of market-oriented innovative subjects, although the proportion of high-tech enterprises and above-scale industrial enterprises with R&D institutions in the Beijing-Tianjin-Hebei region is lower than that in more developed market-oriented areas, such as the Yangtze River Delta and Pearl River Delta, the density of market-oriented innovative subjects in the region is still 3–5 times higher than the national average when considering the fact that the region occupies only 2.27% of the national land area.

In terms of funding and talent, by the end of 2021, the Beijing-Tianjin-Hebei region had gathered 12.45% of the country's scientific and technological innovation talents and 15.3% of its funding expenditures, second only to the Yangtze River Delta region. At the same time, the region had a significant advantage in the proportion of high-end talents, with researchers holding doctoral degrees and academicians of the Chinese Academy of Sciences and the Chinese Academy of Engineering accounting for 22.31% and 37.22% of the country's total, respectively (Table A2). The clustering of high-end innovative talents has strengthened the region's capacity for original and cutting-edge technological innovation, raising the overall level of innovation in the Beijing-Tianjin-Hebei region. In addition, the region's research funding per capita was about CNY 331,200, much higher than the national average of CNY 268,900.

### 4.3. Further Improvement of the Innovation Chain

The high concentration of innovative entities and factors in the Beijing-Tianjin-Hebei region provides a guarantee for the output of scientific and technological achievements, and the gradual improvement of its technology transfer system further improves the innovation chain from output to transformation.

Venture capital is a market mechanism that can accelerate the transformation of scientific and technological achievements and help them make the "critical leap" from laboratory technology to the market [65]. To accelerate the development of this market mechanism, the Beijing-Tianjin-Hebei region has established a complete system of intellectual property and equity trading markets, such as the National Patent Technology (Beijing) Exhibition and Trading Center, the Beijing Property Rights Exchange, the Beijing Equity Exchange, and the National Small and Medium-sized Enterprises Share Transfer System, which facilitate the transfer, transformation, and later exit of innovative achievements, such as patents. In 2021, the Ministry of Finance introduced policies that allow companies and individual angel investors in the Beijing-Tianjin-Hebei region to deduct 70% of their pre-investment amount from their income tax. With multiple favorable policies in place, the number of venture capital cases in the region exceeded 1700 in 2021, accounting for 27.54% of the country's total, with total financing exceeding USD 57 billion, or 35.11% of the national total. At the same time, the financing amount for a single innovative entrepreneurial case was about USD 90 million, which is 3.5 times the national average.

The Beijing-Tianjin-Hebei region also has a relatively complete industrial foundation, which ensures that scientific and technological achievements can be industrialized on a large scale after initial incubation, further magnifying economic benefits. Tianjin is an old northern industrial center and shipping center with a strong industrial foundation and advanced manufacturing and shipping industries. In recent years, Hebei has adjusted and optimized its industrial structure through supply-side structural reforms, with the added value of high-tech industries and equipment manufacturing industries accounting for 18.4% and 27% of the scale of industrial added value, respectively, further strengthening the manufacturing industry foundation.

## 5. Insufficient Construction of the RIE in Beijing-Tianjin-Hebei

Currently, there are challenges to building a RIE in the Beijing-Tianjin-Hebei region, including the seriously uneven distribution of innovation factors, which hinders the overall level of innovation, the leapfrogging of scientific and technological achievements, which affects the long-term stable operation of the system, and the overall low level of innovation among market-oriented innovation entities and the continuous loss of key innovative factors, such as talent.

### 5.1. The Serious Uneven Distribution of Innovation Entities and Factors

The Beijing-Tianjin-Hebei region is one of the areas in China with the most unbalanced distribution of innovative factors. In terms of innovation entities and innovation factors, there is a significant decrease from Beijing to Tianjin to Hebei (Table A3).

Regarding innovation entities, at the end of 2021, the number of scientific research institutions per 10 million people in Beijing was 4.7 times and 17.3 times that of Tianjin and Hebei, respectively. The number of universities was 1.21 times and 2.65 times, and the number of high-tech enterprises was 1.1 times and 4.4 times, respectively. Due to the functional positioning of the cities, Beijing mainly has management-oriented headquarters of enterprises, which are relatively focused on management. The number of research and development institutions established by large-scale enterprises per 10 million people in Beijing was 284.04, which was 54.64% of that of Tianjin, an old industrial base. However, it was still 1.89 times that of Hebei. The gap between the three regions in terms of high-end innovative entities was even more significant. The number of national key laboratories and engineering technology research centers per 10 million people in Beijing was 4.75 times and 83.65 times that of Tianjin and Hebei, respectively, and the "Double First-Class" universities were 4.83 times and 118.7 times that of Tianjin and Hebei, respectively. The gaps in research and development personnel and funding were also significant among the three regions, with Beijing leading Tianjin by a large margin and Tianjin leading Hebei by a large margin. In terms of high-level innovative talents, the gaps between the three regions were stark, with the proportion of research and development personnel with a doctoral degree in Beijing being 4.97 times and 35.33 times that of Tianjin and Hebei, respectively.

Currently, the serious imbalance of resource allocation among Beijing, Tianjin, and Hebei will not only reduce the marginal output of Beijing's innovative factors but also cause the continuous outflow of innovation talents and factors from Tianjin and Hebei to Beijing, as Beijing can provide more innovative jobs, higher innovative platforms, and more superior innovative conditions. This will not only make Beijing an "innovation island" but also impede the construction of a stable innovation ecosystem in the Beijing-Tianjin-Hebei region.

### 5.2. The Innovation Level of Market-Oriented Innovation Entities Needs to Be Improved

The innovation capability of market-oriented innovation entities, such as various types of enterprises in the Beijing-Tianjin-Hebei region, needs to be further improved, as their innovation enthusiasm is low and innovation capability is not high, which will have an adverse impact on the overall innovation efficiency in the region [66].

In terms of the number of innovative entities, in 2021, the number of high-tech enterprises and large-scale enterprises with research and development institutions in the three regions of Beijing-Tianjin-Hebei was only 6.37% and 10.59% of the national total, respectively, which is less than one-fourth of that of the Yangtze River Delta and about half of that of the Pearl River Delta, both regions where market-oriented innovation is more active(Table A4). Regardless of scale, the proportion of enterprises in the Beijing-Tianjin-Hebei region that have established research and development institutions or are engaged in research and development activities is lower than the national average, and the gap with the Yangtze River Delta and the Pearl River Delta, where market-oriented innovation is more active, is even greater. In 2021, the proportion of large-scale enterprises with research and development institutions in the Beijing-Tianjin-Hebei region was only 10.91%,

which is only 66.89%, 34%, and 48% of the national total, the Yangtze River Delta, and the Pearl River Delta, respectively. Among small and medium-sized industrial enterprises, the proportion of innovative companies and the proportion of companies implementing innovation activities in the Beijing-Tianjin-Hebei region are both significantly lower than the national average and those of the Yangtze River Delta and Pearl River Delta regions. In terms of the number of R&D institutions established by each high-tech enterprise, the Beijing-Tianjin-Hebei region is only about 70% of the national average and half that of the Yangtze River Delta and Pearl River Delta regions. Within the three regions of Beijing, Tianjin, and Hebei, the gaps between Beijing and the Yangtze River Delta and Pearl River Delta regions are relatively small, while Hebei Province has a large gap, with huge potential for catching up in the future.

In terms of funding, the science and technology innovation activities in the Beijing-Tianjin-Hebei region are more dependent on government investment. Compared with the more innovative Yangtze River Delta and Pearl River Delta regions, there is still significant room for growth in enterprise investment in R&D in the Beijing-Tianjin-Hebei region. In 2021, government funding accounted for 78.43% of R&D funding in the Beijing-Tianjin-Hebei region, while the figures for the Yangtze River Delta and Pearl River Delta regions were only 18.66% and 8.92%, respectively (Table A5). Looking within the Beijing-Tianjin-Hebei region, Beijing, as a leading area for innovation, had a government-to-enterprise funding ratio of 1.66, indicating the weakness of market-oriented innovation. In addition, since 2009, the ratio of government funding to enterprise funding in Beijing and Tianjin has shown an increasing trend compared to the decreasing trend in the Yangtze River Delta, Pearl River Delta, and Hebei areas, indicating that the growth rate of enterprise R&D investment is lower than that of government investment. In terms of the distribution of innovative talent, in 2021, the number of R&D personnel in large industrial enterprises in the Beijing-Tianjin-Hebei region was about 1.1 times that of universities and research institutions, of which the proportion in Beijing was only about 30%. In contrast, in the Yangtze River Delta and Pearl River Delta regions, the figures were about 5.1 times and eight times, respectively. In summary, compared with the situation in the Yangtze River Delta and Pearl River Delta regions, where enterprises are leading innovation, market-oriented innovation in the Beijing-Tianjin-Hebei region is relatively weak, and there is enormous potential for improvement in innovation level.

*5.3. Obvious Leap Phenomenon in the Process of Technology Transfer*

As the key area of scientific and technological innovation in the Beijing-Tianjin-Hebei region, Beijing not only utilizes its own innovative elements but also uses a large number of innovative factors from Tianjin and Hebei through the "siphon effect". However, the technological achievements produced in Beijing tend to leap over the innovation capabilities in Tianjin and Hebei and are transferred to other regions, which has become increasingly serious. In 2021, the top three destinations for technology transfer from Beijing, in terms of both amount and number of contracts, were local areas, the Yangtze River Delta, and the Pearl River Delta. The amount and number of technology contracts transferred to Tianjin and Hebei were only 3.93% and 5.13%, respectively. Since 2008, the leap phenomenon in technology transfer from Beijing has become increasingly obvious. The proportion of technology contracts transferred to Tianjin and Hebei has decreased from 6.31% to 3.93% in 2020, a decrease of 37.7%; the proportion of the number of contracts has decreased from 5.50% to 5.13%, a decrease of 6.7%. At the same time, the amount and number of contracts transferred to the Yangtze River Delta and the Pearl River Delta have both increased by about 10%.

Beijing has long had an obvious "siphon effect" on the innovation factors of Tianjin and Hebei, which, while improving Beijing's innovation level, is not conducive to the improvement of innovation capabilities in Tianjin and Hebei. The intensification of the leap phenomenon means that after the output of innovation factors from Tianjin and Hebei to Beijing, these regions cannot obtain resources replenished from the converted



achievements, leading to one-way resource consumption and ultimately continuously lowering the innovation level of the two regions. From the perspective of the overall innovation ecosystem, this results in innovation factors in the Beijing-Tianjin-Hebei region failing to form a virtuous cycle, posing a challenge to building a stable and efficient innovation ecosystem.

*5.4. Increasing Pressure on the Outflow of Key Innovation Factors, Such as Talents*

Innovation driving is essentially driven by talent, and talent is the first resource of innovation. In recent years, the Beijing-Tianjin-Hebei region, especially Beijing, has seen a certain degree of outflow of talent, becoming an important challenge for building the RIE.

Taking college graduates, an important source of innovation talents, as an example, in 2021, the proportion of college graduates in Beijing who chose to work in and outside of Beijing was 62.68% and 37.32%, respectively. Compared with the same indicator in 2020, they have decreased and increased by more than 0.6 percentage points, respectively. At the same time, Tianjin and Hebei are not the biggest beneficiaries of the outflow of college graduates from Beijing, as a large number of graduates are flowing to the Pearl River Delta, Hangzhou, Chengdu and other places. By analyzing the employment areas of college graduates from top universities in China, such as Peking University and Tsinghua University, it was found that the proportion of college graduates choosing to work in Beijing has gradually decreased from 58.9% in 2017 to 41.56% in 2021, a decrease of 17.34%. In contrast, the proportion of graduates choosing to work in the Pearl River Delta has risen from 10.23% to 19.89% year by year, with a rising range of 94.43%. In 2021, the proportion of Beijing's college graduates employed in the Tianjin-Hebei region was 5.7%, a decrease of 0.63 percentage points compared to the same period in 2020. Based on the above analysis, college graduates in Beijing are continuing to flow out of the city and across the Tianjin-Hebei region towards other innovation ecosystems, such as the Pearl River Delta.

In 2021, the proportion of Beijing university graduates employed in the Beijing-Tianjin-Hebei region was 5.7%, a decrease of 0.63 percentage points compared to the same period in 2020. Based on the above analysis, Beijing's university graduates are continuously leaving the region and crossing the Beijing-Tianjin-Hebei region to other innovation ecosystems, such as the Pearl River Delta.

The reasons for the talent outflow from the Beijing-Tianjin-Hebei region can be explained by a "push-pull" mechanism. Firstly, the rising cost of living and tightening household registration policies in Beijing in recent years has become important driving forces for talent outflow. This factor has also led to an increase in the proportion of Beijing graduates flowing to Shanghai from 5.16% in 2017 to 9.25% in 2019 and then down to 6.21% in 2021. Secondly, in recent years, there are still significant shortcomings in the development of infrastructure, industrial structure, and environmental conditions, particularly in Hebei province, which has resulted in low attractiveness for talents from Beijing and limits the spillover effect of Beijing's talent resources to these areas. Finally, in recent years, some cities in China have significantly increased their attractiveness to talents by introducing preferential policies for settling, purchasing homes, and optimizing the innovation and entrepreneurship environment, along with lower living costs and good environmental conditions, which have formed a pulling force for talent outflow. With the combined effects of the above push factors in Beijing and multiple pull factors in other regions, the outflow of talent resources in the Beijing-Tianjin-Hebei region has occurred.

## 6. Constructing a RIE in Beijing-Tianjin-Hebei: Based on Its Advantages and Disadvantages

Considering the characteristics of a RIE and the advantages, as well as the disadvantages of the Beijing-Tianjin-Hebei region, the RIE in the Beijing-Tianjin-Hebei region can be constructed as follows.

(1) Use both market and administrative approaches to allocate innovation factors and promote the moderate transfer of innovation factors to the Beijing-Tianjin-Hebei

region. Firstly, rely on the Beijing-Tianjin-Hebei coordinated development leadership group to build a collaborative innovation platform and promote large R&D institutions and universities in Beijing to establish branch institutions or joint research centers in Hebei and Tianjin. Secondly, establish an innovative research personnel exchange mechanism and introduce policies to encourage scientific researchers from Beijing to work part-time or exchange positions with research institutions, universities, and enterprises in Hebei and Tianjin. Thirdly, promote the direct settlement of medical insurance and various forms of recognition of qualifications and benefit-sharing mechanisms for innovation entities and researchers who come to Hebei to protect the rights and convenience of innovation entities and scientific researchers.

(2) Stimulate the innovation vitality of market-oriented innovation entities and improve the innovation level. Currently, the technology innovation investment and innovation level of market-oriented innovation entities in the Beijing-Tianjin-Hebei region is relatively low, which will generally reduce the region's innovation efficiency and the application orientation of innovation. To address this issue, first, the three regions should adopt policies that treat all market-oriented innovation entities equally with government-led innovation entities in terms of funding, project approval, and treatment of scientific and technological talents. Secondly, leverage the government's funding leverage mobilizes social capital to support industrial innovation. The governments of the three regions can jointly or separately invest in venture capital as subordinate funds and invest in enterprises through market-oriented means to support the innovation and transformation activities of market-oriented innovation entities.

(3) Multiple measures should be taken to enhance Hebei Province's capacity to undertake technological transfers from Beijing and Tianjin. In order to effectively promote the transfer and commercialization of technological achievements from Beijing and Tianjin, Hebei Province has implemented various policies, such as tax incentives, financial subsidies, and discounted loans, to support companies that meet certain requirements for research and development investment or output. It has also increased the protection of intellectual property rights to ensure the legitimate rights and interests of innovative enterprises and enhance their enthusiasm for research and development investment.

Hebei Province has focused on promoting the construction of entrepreneurship parks and industrial parks in the Beijing New Airport Economic Zone, Caofeidian, Bohai New Area, Zhengding New Area, and other areas around Beijing and Tianjin, taking advantage of the geographic proximity to undertake technological transfers and commercialize achievements from these two cities. After the construction of Xiong'an New Area and Beijing's sub-center is completed, Hebei Province will continue to promote its capacity to undertake technological transfers and commercialize achievements from these two regions.

Fundamentally, Hebei Province still needs to free up its thinking and promote the transformation of the government from a management-oriented model to a service-oriented one. It needs to strictly enforce a list of powers and prevent any incidents of corruption. This will effectively improve the local business environment. At the same time, Hebei Province should accelerate the integration with Beijing and Tianjin in terms of tax, registration, and other mechanisms and explore ways to attract technology enterprises from Beijing and Tianjin by retaining their original registered names and only changing their addresses.

(4) Build an innovative environment that is livable and business-friendly and achieve the internal circulation of talent resources in the Beijing-Tianjin-Hebei region. Research shows that talent mobility is generally characterized by high education and age under 35. The willingness to move is influenced by many factors, such as income level, personal development potential, living environment, children's education, medical care, and comprehensive living costs. To reduce the outflow of talent resources, efforts can be made from two aspects: enhancing the attractiveness of talent and promoting cross-regional commuting in the Beijing-Tianjin-Hebei region.

On the one hand, Beijing and Tianjin can accelerate the implementation of point-based household registration, provide certain housing subsidies for renting near work or living,

and enhance the attractiveness of innovative talents, especially high-end innovative talents. Hebei Province can consider lowering the threshold for household registration, simplifying registration procedures, further improving environmental pollution control, providing certain housing purchase subsidies or tax incentives to talents, strengthening its industrial foundation, expanding its employment capacity while reducing its own outflow of talent and improving its capacity to undertake talent spillovers from Beijing.

On the other hand, the three regions of Beijing, Tianjin, and Hebei can improve the interconnection of infrastructure to facilitate cross-regional commuting and further reduce the outflow of talent. Hebei Province should accelerate the construction of high-speed railways and highways that connect directly to Beijing and promote the direct connection of the Beijing Tianjin subway to Hebei Province. It should also increase the density of direct buses and subways between the surrounding areas of Beijing and Tianjin. Meanwhile, Tianjin should increase the frequency and adjust the operating times of the Beijing-Tianjin intercity high-speed railway to provide convenience for cross-regional commuting.

## 7. Discussion

Regional innovation ecosystems are crucial for fostering innovation, economic growth, and competitiveness. While both developed and developing countries strive to build such ecosystems; they face distinct challenges and opportunities due to differences in their contextual factors, institutional capacities, and resources. The first aspect is contextual factors. The literature demonstrates that contextual factors, such as culture, history, infrastructure, and local resources, significantly influence the development of regional innovation ecosystems [67,68]. Developed countries often possess advanced infrastructure, mature institutional structures, and a history of innovation, which can facilitate the creation and growth of innovation ecosystems. In contrast, developing countries may lack such enabling environments, hindering the development of their innovation ecosystems. The second aspect is stakeholder collaboration. Stakeholder collaboration is vital for building robust regional innovation ecosystems. The Triple Helix Model, emphasizing university-industry-government linkages, is widely recognized as a key driver of innovation and economic growth [69]. Developed countries typically exhibit stronger collaborations among these stakeholders due to established networks, resources, and institutional capacities. In contrast, developing countries may face challenges in forging and maintaining such linkages due to weaker institutions, limited resources, and less advanced networks. The third aspect is about policy interventions. Governments play a critical role in supporting regional innovation ecosystems through various policy interventions, such as funding research and development, promoting technology transfer, and implementing regulatory frameworks [70,71]. Developed countries often have more extensive and targeted policies, reflecting their advanced institutional capacities and resources. In contrast, developing countries may struggle to implement effective policy interventions due to limited resources, capacity constraints, and competing policy priorities. The last aspect is about challenges and opportunities. Both developed, and developing countries face unique challenges and opportunities in constructing regional innovation ecosystems. Developed countries may grapple with issues such as maintaining competitiveness, adapting to disruptive technologies, and fostering sustainable growth. On the other hand, developing countries often contend with limited human capital, inadequate infrastructure, weak intellectual property protection, and a lack of financial resources [72,73].

Although the existing literature has conducted a relatively in-depth investigation into the issues faced by developed and developing countries in building regional innovation ecosystems, there is limited research specifically addressing countries like China, which is currently transitioning from a developing to a middle-income country. This represents a significant divergence from and a valuable contribution to the current body of research.

As with the methodology employed to conduct the related study, the existing study focuses primarily on pure theoretical research, with less attention paid to real-world cases. Adner and Kapoor (2010) [74] investigated the impact of the structure of technological inter-

dependence on firm performance within innovation ecosystems. They aimed to understand the value creation process in these ecosystems by employing a combination of theoretical modeling and empirical analysis. By examining the optical disc industry as a case study, the authors found that the structure of technological interdependence significantly influenced firm performance. Firms that effectively managed their interdependencies were better positioned to capture value from their innovations. Consequently, the study proposed a framework for understanding the value creation process in innovation ecosystems, emphasizing the need for strategic management of interdependencies. Dedehayir and Steinert (2016) [75] focused on providing a comprehensive review of the hype cycle model, a popular method used to analyze the dynamics of innovation ecosystems. The authors conducted a systematic literature review of the hype cycle model and its applications, evaluating its strengths, limitations, and potential improvements. The study identified opportunities for refining and extending the hype cycle model, especially by integrating external factors, such as social, political, and environmental influences, into the analysis of innovation ecosystems. The authors suggested that incorporating these factors into the hype cycle model would yield a more comprehensive understanding of innovation ecosystem dynamics, which could inform the construction and management of such ecosystems more effectively. Clarysse et al. (2014) [76] explored the process of value creation in ecosystems, with a particular focus on the transition from knowledge ecosystems to business ecosystems. The authors employed a multiple case study approach, examining four university spin-off companies that successfully transitioned from knowledge ecosystems to business ecosystems. The study identified several factors that facilitated this successful transition, such as the involvement of experienced entrepreneurs, the establishment of strategic partnerships, and the development of complementary assets. The authors proposed a framework for understanding the value creation process in ecosystems and highlighted the importance of managing the transition between knowledge and business ecosystems for successful innovation. In conclusion, these three representative papers contribute valuable insights to the understanding of constructing innovation ecosystems by examining various aspects of value creation, ecosystem dynamics, and the transition between different types of ecosystems. Compared with the above papers, this article has a unique perspective. This article first systematically compares the RIE with the natural ecosystem and, by analyzing the characteristics of the natural ecosystem, summarizes the characteristics of the regional innovation ecosystem. Taking this as a guide, based on the actual situation of Beijing-Tianjin-Hebei, how to build a RIE, in reality, was systematically explored. From this perspective, this article achieves the integration of theory and practice, which is the most significant difference between this article and the aforementioned articles.

## 8. Conclusions and Limitations

### 8.1. Conclusions

Based on the inspiration of the most efficient natural ecosystem, the tropical rainforest ecosystem, the characteristics of RIE were summarized as creating an innovation ecosystem environment conducive to the development and growth of innovation activities, multi-level, differentiated producers, consumers, and decomposers of innovation activities and innovation factors realizing a virtuous closed-loop cycle within the innovation ecosystem. To explore how to construct a RIE According to these characteristics, this paper employed the Beijing-Tianjin-Hebei region in China as the case and systematically investigates the advantages of Beijing-Tianjin-Hebei region in constructing a RIE of the innovation environment, the further improvement of innovation resource aggregation and innovation chains, as well as the disadvantages of an unbalanced distribution of innovation resources, the leapfrogging transformation of scientific and technological achievements, low innovation levels of market-oriented innovation entities, and loss of innovative factors, such as talents. On account of the advantages and disadvantages of the Beijing-Tianjin-Hebei region above, this paper put forward the suggestions of constructing a RIE in this region that uses both market and administrative approaches to allocate innovation factors and promote the mod-

erate transfer of innovation factors, stimulating the innovation vitality of market-oriented innovation entities, taking multiple measures to enhance Hebei Province's capacity to undertake technological transfers from Beijing and Tianjin and building an innovative environment that is livable and business-friendly and achieve the internal circulation of talent resources.

### 8.2. Limitations

This paper, grounded in relevant theories of RIE, takes the Beijing-Tianjin-Hebei region in China as a case study to systematically explore the construction of RIE in reality. Overall, there are two notable limitations in this study. First, there is a lack of quantitative analysis. The structure and characteristics of RIE, as well as the construction of the Beijing-Tianjin-Hebei RIE, lack quantitative model support, which increases the difficulty for other researchers to apply the findings to other regions. Second, the study lacks a cross-regional comparison between the Beijing-Tianjin-Hebei RIE and those of other countries, including both developed and developing countries, which prevents a comprehensive presentation of the strengths and weaknesses of the Beijing-Tianjin-Hebei RIE. In the future, based on existing research, we will further support the research process and results with quantitative models and conduct cross-regional comparisons between China, a country transitioning from a developing to a developed nation, and similar cases in both developed and developing countries, which will enable us to explore the regional heterogeneity in constructing RIE.

**Author Contributions:** Writing—original draft preparation, Y.C. and H.M.; writing—review and editing, Y.C., J.L., Y.Y., X.L., Z.L., N.L., Z.W. and H.L.; supervision, H.M.; project administration, Y.C. All authors have read and agreed to the published version of the manuscript.

**Funding:** This research was funded by the National Natural Science Foundation of China, "A Study on the Co-evolution of the Relationship between the Power Pattern and the Geo-economy in the Arctic Region from the Perspective of Evolutionary Game", grant number 42201243.

**Institutional Review Board Statement:** Not applicable.

**Informed Consent Statement:** Informed consent was obtained from all subjects involved in the study.

**Data Availability Statement:** Not applicable.

**Conflicts of Interest:** The authors declare no conflict of interest.

## Appendix A

**Table A1.** Proportions of various types of innovative entities in Beijing-Tianjin-Hebei and other regions in China in 2021 (%).

| Regions | Universities | First-Class Universities | First-Class Disciplines | R&D Institutions | National Key Laboratories | National Engineering and Technology Research Centers | High-Tech Enterprises | Large-Scale Enterprises with Established R&D Institutions |
|---|---|---|---|---|---|---|---|---|
| Beijing-Tianjin-Hebei | 10.25 | 23.81 | 48.97 | 14.87 | 33.86 | 22.22 | 6.37 | 10.59 |
| Yangtze River Delta | 12.98 | 16.67 | 35.40 | 10.25 | 24.02 | 18.06 | 27.90 | 50.01 |
| Pearl River Delta | 5.66 | 4.76 | 5.31 | 5.59 | 4.33 | 6.39 | 21.33 | 15.70 |

Data source: Ministry of Education, Ministry of Science and Technology, and Science and Technology Statistical Yearbook of China in 2022.

**Table A2.** Proportions of innovative factors in Beijing-Tianjin-Hebei and other regions in China in 2021 (%).

| Regions | R&D Staffs | R&D staffs with Doctoral Degrees | Academicians | R&D Expenditure |
|---|---|---|---|---|
| Beijing-Tianjin-Hebei | 12.45 | 22.31 | 37.22 | 15.3 |
| Yangtze River Delta | 26.28 | 21.34 | 14.6 | 26.83 |
| Pearl River Delta | 12.61 | 7.5 | 2.1 | 12.98 |

Data source: Science and Technology Statistical Yearbook of China in 2022' Chinese Academy of Sciences, and Chinese Academy of Engineering. Academicians refer to members of the Chinese Academy of Sciences and the Chinese Academy of Engineering.

**Table A3.** Distribution of various types of innovation entities and innovation resources in the Beijing-Tianjin-Hebei region in 2021.

| Region | Research Institutions | Key Research Institutions * | Universities | Double First-Class Universities | High-Tech Enterprises | Large-Scale Enterprises with Established R&D Institutions | R&D Stuffs | R&D Staffs with Doctoral Degrees | Total R&D Expenditure | Per Capita Expenditure on R&D Staff |
|---|---|---|---|---|---|---|---|---|---|---|
| Beijing | 185.31 | 66.92 | 42.58 | 15.44 | 372.02 | 284.04 | 17.47 | 3.18 | 1484.6 | 39.76 |
| Tianjin | 39.05 | 10.88 | 35.21 | 3.20 | 341.23 | 519.85 | 11.34 | 0.64 | 537.32 | 30.32 |
| Hebei | 10.71 | 0.80 | 16.06 | 0.13 | 84.74 | 150.33 | 2.35 | 0.09 | 383.4 | 21.84 |

* Key research institutions refer to national key laboratories and engineering technology research centers. The units for research institutions, national key laboratories and engineering technology research centers, number of universities, 'Double First-Class' universities, high-tech enterprises, and large-scale enterprises with established R&D institutions are in units of number per ten million people; the units for R&D personnel and R&D personnel with doctoral degrees are in units of number per thousand people; the unit for total expenditure on R&D is in hundred million yuan; the unit for per capita expenditure on R&D personnel is in ten thousand yuan. Data source: Comprehensive statistics from the Statistical Yearbook of China in 2022, Science and Technology Statistical Yearbook of China in 2022, 2021 Annual Report of National Key Laboratories, 2021 Annual Report of National Engineering Technology Research Centers, and the list of universities and disciplines under 'Double First-Class' construction by the Ministry of Education.

**Table A4.** Innovation activities of various types of enterprises in Beijing-Tianjin-Hebei and other regions in China in 2021 (%, PCS).

| Region | Large-Scale Industrial Enterprises | | Small and Medium-Sized Industrial Enterprises | | The Average Number of R&D Institutions Per High-Tech Enterprise |
|---|---|---|---|---|---|
| | Proportion of Enterprises with Established R&D Institutions | Proportion of Enterprises with R&D Activities | Proportion of Enterprises with R&D Activities | Proportion of Enterprises that Have Achieved Innovative Outputs | |
| China | 16.31 | 22.95 | 19.40 | 18.80 | 0.45 |
| Beijing-Tianjin-Hebei | 32.05 | 37.00 | 20.57 | 20.22 | 0.63 |
| Yangtze River Delta | 22.71 | 25.60 | 30.50 | 29.00 | 0.57 |
| Pearl River Delta | 10.91 | 21.06 | 17.62 | 17.08 | 0.31 |
| Beijing | 18.17 | 34.58 | 19.00 | 17.90 | 0.37 |
| Tianjin | 15.62 | 39.46 | 19.60 | 19.30 | 0.25 |
| Hebei | 7.61 | 11.52 | 14.50 | 14.40 | 0.29 |

Data source: Science and Technology Statistical Yearbook of China in 2022.

**Table A5.** Proportions of government funding and enterprise funding in Beijing-Tianjin-Hebei and other regions in China from 2014 to 2021 (%).

| Region | 2014 | 2015 | 2016 | 2017 | 2018 | 2019 | 2020 | 2021 |
|---|---|---|---|---|---|---|---|---|
| Yangtze River Delta | 19.81 | 20.85 | 18.96 | 19.71 | 18.16 | 18.00 | 18.92 | 18.66 |
| Pearl River Delta | 9.91 | 9.28 | 10.32 | 9.89 | 9.00 | 8.07 | 9.08 | 8.92 |
| Beijing-Tianjin-Hebei | 85.90 | 96.62 | 80.14 | 77.45 | 81.73 | 76.78 | 83.85 | 78.43 |
| Beijing | 144.92 | 174.57 | 154.15 | 153.54 | 169.41 | 161.05 | 167.64 | 165.57 |
| Tianjin | 23.76 | 25.94 | 20.56 | 20.44 | 22.05 | 20.38 | 27.89 | 26.22 |
| Hebei | 28.92 | 22.45 | 19.47 | 18.99 | 16.35 | 16.18 | 18.80 | 18.07 |

Data source: Science and Technology Statistical Yearbooks of China in various years.

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
