# Peer review of "Construct a Regional Innovation Ecosystem: A Case Study of the Beijing-Tianjin-Hebei Region in China"

_sustainability, doi:10.3390/su15097099_

Round 1

Reviewer 1 Report

Thank you for possibilty of reviewing paper titled "How to Construct the Regional Innovation Ecosystem? A case study of Beijing-Tianjin-Hebei Region in China" It could be interesting, but needs some major improvements.

- The goal of the paper does not flow from previous studies. Also authors should show what new they introduced to existing knowledge. 

I suggest to higlight position of Chinese economy in Introduction. Chinese cities have raising their economic intercity (especially international) connectivities (https://doi.org/10.3390/land11091574), also their companies raising their economic power (DOI: 10.3390/su132212798), and Beijing is one of the leading cities in Information Technology sector in the world (doi: 10.1007/s11769-017-0850-5)

- Methods chapter doesn't contain knowledge how RIE was created/ calculated. The method shown should be clear and available to copy by other persons. Math formulas, etc. needed

- Details about rainforest (chapter 3.1) are not necessary. The paper is about Bejing-... not about rain forests from southern China. I'm also confused with 3.2. What is the connection between rainforest ecosystem and Bejing-... ecosystem? Why it's important to spend so many words for rainforest RIE? - especially they didn't mentioned about rainforest RIE in results, and conclusions. Authors should delete 3.2 or highlight similarities or things/policies, that could be transfered from rainforest to their study area.

- Discusson part is needed. In this part authors should show main findings, and compare them with previous studies

Author Response

Thank you for possibilty of reviewing paper titled "How to Construct the Regional Innovation Ecosystem? A case study of Beijing-Tianjin-Hebei Region in China" It could be interesting, but needs some major improvements.

 A:Thank you for your kind acceptance. We will try our best to revise the manuscript.

- The goal of the paper does not flow from previous studies. Also authors should show what new they introduced to existing knowledge. 

I suggest to higlight position of Chinese economy in Introduction. Chinese cities have raising their economic intercity (especially international) connectivities (https://doi.org/10.3390/land11091574), also their companies raising their economic power (DOI: 10.3390/su132212798), and Beijing is one of the leading cities in Information Technology sector in the world (doi: 10.1007/s11769-017-0850-5)

A:Thank you for your kind suggestion. Based on your suggestions, and taking into account the recommendations from other peer reviewers, we have made substantial revisions to the Introduction section. Please check line 50-88.

- Methods chapter doesn't contain knowledge how RIE was created/ calculated. The method shown should be clear and available to copy by other persons. Math formulas, etc. needed

A:Thank you for your kind suggestion.We have made significant revisions to the Methods section. However, this paper does not involve mathematical models, so we did not add mathematical formulas, etc. In addition to structural adjustments, we have also included sections on the study area and data sources. Please check line 99-119.

- Details about rainforest (chapter 3.1) are not necessary. The paper is about Bejing-... not about rain forests from southern China. I'm also confused with 3.2. What is the connection between rainforest ecosystem and Bejing-... ecosystem? Why it's important to spend so many words for rainforest RIE? - especially they didn't mentioned about rainforest RIE in results, and conclusions. Authors should delete 3.2 or highlight similarities or things/policies, that could be transfered from rainforest to their study area.

A:We think that there is a close correlation between RIE and natural ecosystems. In fact, the former is actually obtained from what the latter gives. Therefore, in order to analyze the characteristics of RIE, an effective approach is to systematically analyze the characteristics of natural ecosystems. For this reason, we did not delete 3.2 and supplemented the internal connection between the concepts of innovation ecosystem and natural ecosystem. Please Check line 126-152.

- Discusson part is needed. In this part authors should show main findings, and compare them with previous studies

A:Thank you for your kind guidance. Follow your suggestion, the discussion part is added. Please check line 598-672.

Author Response

Dear reviewer,

Thank you for your kind acceptance. We have tried our best to revise the manuscript as follows.

  1. As with the title of the paper, we have made revision according to your suggestion.
  2. As with the ABSTRACT, based on the 7 specific comments you provided, and taking into account the opinions of other peer reviewers, we have made extensive revisions to the Abstract section. Please check lines 10-30.
  3. As with the Keywords, we have deleted one keyword “tropical rainforest”. Please check line 31.
  4. As with the Introduction section, based on the 7 specific comments you provided, and taking into account the opinions of other peer reviewers, we have made extensive revisions. Please check line 53-68 and 77-88.
  5. As with the section ofMethodology, we have made extensive revisions. Firstly, we have made minor adjustments to the comparative analysis section; in addition, we have made corresponding adjustments to the original second paragraph that you suggested we modify; also, the text you recommended to remove has been deleted accordingly.
  6. Regarding your revision suggestions for "The Basic Features of RIE: Insights from Tropical Rainforest," we have generally accepted your opinions and made corresponding changes. It is worth noting that for Figure 1 and Figure 2, which you suggested to modify, we have deleted them based on your advice and the recommendations of other peer reviewers. We have also added a comparison table between the regional innovation ecosystem and the natural ecosystem. Please review the changes we have made.

  1. Regarding your revision suggestions for the "Conditions for constructing a RIE in a combined region of Beijing-Tianjin-Hebei" section, we first modified the title of this part; secondly, we believe that this section is closely related to the whole text, including the title. This is because our aim is to use the Beijing-Tianjin-Hebei region as an example to explore how to construct a regional innovation ecosystem. However, the basic premise for constructing a regional innovation ecosystem is to conduct an in-depth discussion on the strengths and weaknesses of the region, and then, based on this, carry out the subsequent research on how to construct a regional innovation ecosystem. Therefore, we did not make significant changes to the structure of this section.

  1. Regarding your revision suggestions for the “Insufficient construction of the RIE in Beijing-Tianjin-Hebei”, our considerations for the revisions are the same as mentioned above.

9.Thank you for your professional suggestions, based on which we have made detailed revisions. At the same time, in combination with the opinions of other peer reviewers, we have adjusted the structure of the article. The main change is that we have moved the policy recommendations proposed in the original text to Section 6, which is the discussion on how to construct a regional innovation ecosystem in the Beijing-Tianjin-Hebei region.

Reviewer 3 Report

Article title: How to Construct the Regional Innovation Ecosystem? A case study of Beijing-Tianjin-Hebei Region in China

This research seems very interesting, but I have some comments about improving this paper.

Abstract

- Academic research/knowledge gap, methodology, major findings, and policy implications need to be added.

1. Introduction

- Studies from developed and developing countries need to be added.

- The research/knowledge gap should be clearly mentioned.

- The organization of the manuscript needs to be added. 

2. Methodology

- Add rationale for selecting study area and spatial map. 

- Characteristics of the study area are missing. 

- Data and data sources need to be explained. 

- The methodology part should be concrete and comprehensive. 

- Line 81 - 88: It is better to add in introduction part as the organization of the manuscript.

3. The Basic Features of RIE: Insights from Tropical Rainforest

- Table 1: Add reference(s).

- Figure 1 & 2: Add reference(s).

- Line 173 - 189: References are missing.

4. Favorable conditions for constructing a RIE in Beijing-Tianjin-Hebei

- Section 4 (Line 217 - 312): It seems that this comes from the previous literature/documents. Please add the proper references if applicable.

5. Insufficient construction of the RIE in Beijing-Tianjin-Hebei

- I suggest adding a comparative matrix for a clear understanding of the readers.

- Add discussion confronts from developed and developing countries.

6. Conclusions and Policy Suggestions

- Limitations and future research directions are missing.

Author Response

Article title: How to Construct the Regional Innovation Ecosystem? A case study of Beijing-Tianjin-Hebei Region in China

This research seems very interesting, but I have some comments about improving this paper.

 A:Thank you for your kind acceptance. We will try our best to revise the manuscript.

Abstract

- Academic research/knowledge gap, methodology, major findings, and policy implications need to be added.

 A:Thank you for your kind guidance, which has helped us adding related words in the abstract part. Please check line 10-15 and 25-28.

  1. Introduction

- Studies from developed and developing countries need to be added.

- The research/knowledge gap should be clearly mentioned.

- The organization of the manuscript needs to be added. 

  A:Thank you for your kind guidance, which has helped us adding related words in this part.

  1. Methodology

- Add rationale for selecting study area and spatial map. 

- Characteristics of the study area are missing. 

- Data and data sources need to be explained. 

- The methodology part should be concrete and comprehensive. 

- Line 81 - 88: It is better to add in introduction part as the organization of the manuscript.

 A:Thank you for your kind guidance, which has helped us adding related words in the abstract part.

  1. The Basic Features of RIE: Insights from Tropical Rainforest

- Table 1: Add reference(s).

- Figure 1 & 2: Add reference(s).

- Line 173 - 189: References are missing. 

A:Thank you for your kind guidance.It should be noted that in order to ensure the compactness of the article and in combination with the revision suggestions proposed by other peer reviewers, we have deleted the original Table 1 and Figures 1-2. In addition, lines 173-189 did not miss any references, as this section originally did not involve any references.

  1. Favorable conditions for constructing a RIE in Beijing-Tianjin-Hebei

- Section 4 (Line 217 - 312): It seems that this comes from the previous literature/documents. Please add the proper references if applicable.

A: This is the author's interpretation of the data, and it does not involve any references.

  1. Insufficient construction of the RIE in Beijing-Tianjin-Hebei

- I suggest adding a comparative matrix for a clear understanding of the readers.

- Add discussion confronts from developed and developing countries.

A:Thank you for your suggestion. We believe that matrices are not conducive to reader comprehension, so our approach is to create separate tables. Due to space constraints, we have placed them in the appendix. Moreover, based on your recommendation to add related discussions, we have included additional discussion accordingly. Please see lines 598-672.

  1. Conclusions and Policy Suggestions

- Limitations and future research directions are missing.

A: We have added this part, please check line 695-709.

Reviewer 4 Report

The general merit of the research presented in the article is good. The exciting topic, however, needs more appropriate content to benefit readers who need insight into the issue. Regional Innovation Ecosystem (RIE) should be discussed in the context of its impact on the involved cities and regions. Moreover, other examples of RIE should be mentioned here - for example, once from UE and Canada. Moreover, the objectives of the study should be presented more precisely. The abstract and introduction need to provide the aims or objectives of the study in the way they could be evaluated, more over the presented method does not give the possibility to do so. The method focuses on different types of analysis and needs better support with data sources and general research process explanations. A graphical representation of the process would be a good addition here. Moreover, based on the applied method, the result should address the objectives in a structured way. The Abstract describes the topic clearly but would benefit from a more structured approach- with clear sections on Background/theoretical framework, Methods, Results, and Conclusions. The introduction provides the data on the study's general idea but needs a sufficient theoretical framework. The topic has been discussed in dozens of publications. The article should also include a discussion with a broader view of the possible impact of research output with some references to the other regions that applied this strategy or are willing to do so in the future.

Author Response

Thank you for your professional review suggestions, based on which we have made systematic revisions. Specifically, they are as follows:

  1. Regarding "Regional Innovation Ecosystem (RIE) should be discussed in the context of its impact on the involved cities and regions. Moreover, other examples of RIE should be mentioned here - for example, once from EU and Canada.", we have made corresponding supplements, please checklines 50-59;

  1. Regarding "the objectives of the study should be presented more precisely.", we have further modified and clarified the research objectives of the article, please check lines 60-65;

  1. Regarding "The abstract and introduction need to provide the aims or objectives of the study in the way they could be evaluated, moreover the presented method does not give the possibility to do so.", we have also made corresponding changes, please checklines 10-15 and 60-65;

  1. For the research method section, based on the review opinions of you and other experts, we have adjusted the structure of this section, supplemented the research area and data sources, and added Figure 1 to illustrate the entire research process;

  1. Regarding "A graphical representation of the process would be a good addition here", we have accepted your suggestion and created Figure 1;

  1. Regarding "based on the applied method, the result should address the objectives in a structured way.", based on the opinions of you and other peer reviewers, we have adjusted the structure of the article, with the overall approach as follows: â‘ What are the characteristics of the regional innovation ecosystem? â‘¡Taking the Beijing-Tianjin-Hebei region as a case study, what are the strengths and weaknesses of this region in building a regional innovation ecosystem with the aforementioned characteristics? â‘¢Based on these strengths and weaknesses, how should we construct the Beijing-Tianjin-Hebei regional innovation ecosystem?

  1. Based on the suggestions from you and other peer reviewers on the abstract, we have made significant adjustments to the abstract section;

  1. For the suggestion of further discussion, we have supplemented the discussion section.

Round 2

Reviewer 1 Report

Authors have made big effort to improve the paper, however they ignored my suggestions about general information about Chinese economy

(I suggest to higlight position of Chinese economy in Introduction. Chinese cities have raising their economic intercity (especially international) connectivities (https://doi.org/10.3390/land11091574), also their companies raising their economic power (DOI: 10.3390/su132212798), and Beijing is one of the leading cities in Information Technology sector in the world (doi: 10.1007/s11769-017-0850-5)

This part must be strengtened by suggested references.

Author Response

Thanks for your kind acceptance for our previous revision. As with the general information about Chinese economy, we have added related words according to your suggested references, Please check line 70-78.

Thank you for your suggestion again.

Reviewer 3 Report

The authors incorporated the comments. 

Author Response

Thank you for your kind acceptance.

Reviewer 4 Report

Thank you for addressing all the remarks. After the review, the paper gives a better inside into the topic and provides a good theoretical framework. The overall merit is acceptable. However, there are still minor mistakes in the English language and style. 

Author Response

Thank you for your kind acceptance about our revision. Based on your suggestion about the English language, we made a thorough revision. 

Round 3

Reviewer 1 Report

I accept in present form